# Position: Virtual Cells Need Context, Not Just Scale

Payam Dibaeinia [1]   Sudarshan Babu [1]   Mei Knudson [1 2]   Ali ElSheikh [3]   Yibo Wen [3]
Han Liu [3]   Jason Perera [1]   Aly A. Khan [1 2]

## Abstract

The intersection of AI and biology has entered a phase of explosive growth, driven by the ambition to build "Virtual Cells" or computational models capable of predicting cellular responses to any perturbation. Following the success of structural biology (e.g., AlphaFold) and large language models, the field has converged on training massive, high-capacity models on large-scale single-cell data. This position paper argues that scaling model capacity is insufficient to solve the Virtual Cell problem because the primary failure mode is a *lack of adequate coverage over diverse biological contexts*, not insufficient model expressivity. We support this claim by reviewing recent studies showing that simple baselines perform on par with sophisticated architectures within a given biological context, and current models fail to consistently generalize across contexts. We connect this finding to the causal inference literature on transportability and contrast it with domains where scaling has succeeded. We substantiate our argument through analysis of a state-of-the-art model on a 22-million-cell immunology dataset. We conclude that the community faces a *causal transport problem* that cannot be solved by accumulating more data from the same distributions. Instead, we argue that contextual diversity and causal representation learning deserve increased emphasis, complementing ongoing scaling of model capacity and data volume.

## 1. Introduction

AI and biology represent a frontier of considerable excitement, where success promises breakthroughs in understanding the human condition, treating disease, and programming cellular behavior. Following the transformative success of AlphaFold in protein structure prediction (Jumper et al., 2021) and GPT-class models in language, the machine learning community has turned its attention to the "Virtual Cell." In current usage, a Virtual Cell is a model that predicts the perturbed molecular state of a cell (most commonly its mRNA expression profile) given the unperturbed expression profile, an explicit perturbation identifier (e.g., gene knockdown, small molecule), and the cell's biological context, such as donor identity, cell type, tissue of origin, and stimulation timepoint (Bunne et al., 2024).

A reliable Virtual Cell would dramatically accelerate drug discovery, enable personalized medicine at scale, and provide mechanistic insight into diseases that have resisted decades of investigation. Major initiatives spanning academia, industry, and philanthropy have coalesced around this vision (Bunne et al., 2024; Gandhi et al., 2025). Yet a troubling pattern has emerged from recent benchmarks: despite their computational expense and architectural sophistication, cell "foundation models" and deep learning approaches often fail to significantly outperform simple linear baselines (Ahlmann-Eltze et al., 2025; Viñas Torné et al., 2025). This raises a fundamental question: do these failures stem from insufficient data scale and model capacity, or from deeper limitations in how the problem is currently framed?

**We take the position that current predictive models of cellular responses fail primarily due to insufficient coverage of biological contexts, rather than limitations in data volume or model capacity.** The field has been seduced by the allure of an "AlphaFold moment" without recognizing that cellular response prediction is a fundamentally different problem class. Protein folding obeys relatively context-invariant physical laws: the same sequence folds to the same structure regardless of whether the protein resides in a T cell or a neuron (Anfinsen, 1973). Cellular response, by contrast, is exquisitely context-dependent. The same genetic perturbation produces dramatically different phenotypes depending on cell type, activation state, microenvironment, and epigenetic history (Figure 1) (Nadig et al., 2025; Frangieh et al., 2021).

This is not a new lesson. Statistical genetics established

[1]Biohub, Chicago, IL, USA [2]University of Chicago, Chicago, IL, USA [3]Northwestern University, Evanston, IL, USA. Correspondence to: Aly A. Khan <aakhan@uchicago.edu>.

*Proceedings of the $43^{rd}$ International Conference on Machine Learning*, Seoul, South Korea. PMLR 306, 2026. Copyright 2026 by the author(s).

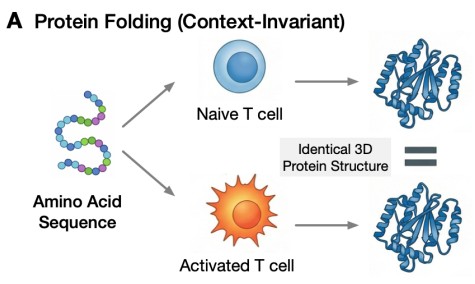

*Figure 1.* **The fundamental distinction between protein structure prediction and cellular response prediction.** **(A)** Protein folding is governed by context-invariant physics – the same amino acid sequence folds to the same structure whether in a T cell, neuron, or test tube. **(B)** Cellular response is context-dependent:the same genetic perturbation (e.g., knockdown of gene X) produces different transcriptional outcomes depending on the cell's activation state, microenvironment, and epigenetic history.

decades ago that genetic effects on gene expression are profoundly tissue- and context-specific (GTEx Consortium, 2020; Stranger et al., 2012). A variant that strongly influences a gene's expression in a T cell may have no effect in brain. The perturbation prediction field is rediscovering what statistical genetics has long known: *context matters*.

Whether the perturbation-to-response mapping is context-invariant or context-dependent has profound implications. If the mapping from perturbation to response is context-invariant, scaling data and model capacity is reasonable because we should learn a fixed function with increasing fidelity. But if the mapping itself varies with context, then accumulating observations from the same contexts yields diminishing returns, shifting models from learning generalizable structure toward memorizing context-specific correlations.

**Contributions.** In this paper we (1) formalize the distinction between context-invariant and context-dependent prediction, connecting to the causal transportability literature and explaining why standard distribution shift frameworks are insufficient; (2) present empirical evidence from published benchmarks and new experiments on a 22M-cell T cell dataset showing that context diversity, *not only cell count*, drives cross-context generalization; and (3) propose concrete recommendations for data generation, benchmarking, and Virtual Cell modeling.

## 2. Related Work

**Scaling paradigm in single-cell biology.** Following successes in NLP and structural biology, the field has pursued a scaling approach to perturbation prediction: pretrain high-capacity models on large observational corpora, then fine-tune or prompt for downstream tasks. Foundation models like scBERT (Yang et al., 2022), Geneformer (Theodoris et al., 2023) and UCE (Rosen et al., 2023) learn representations via BERT-style reconstruction, while autoregressive

approaches like scGPT (Cui et al., 2024) and Transcript-Former (Pearce et al., 2025) generate gene-count tokens iteratively. Perturbation-specific methods like CPA (Lotfollahi et al., 2023) and GEARS (Roohani et al., 2022) train directly on interventional data (sometimes alongside pretrained representations). Appendix A briefly reviews several models. The shared premise is that sufficient scale will internalize the regulatory logic needed for generalization.

**Empirical challenges to the paradigm.** Recent systematic evaluations challenge this premise. Ahlmann-Eltze et al. (2025) systematically compared deep learning and foundation models against linear baselines, finding that none consistently outperformed simple approaches. Viñas Torné et al. (2025) showed that commonly used evaluation metrics can be driven by *systematic variation*, correlated expression changes across perturbations that allow models to score well by predicting average responses rather than perturbation-specific effects. The 2025 Virtual Cell Challenge likewise found no model dominated across metrics, introducing a "Generalist Prize" to recognize robust performances (Arc Institute, 2025).

Importantly, these limitations are not unique to transcriptomic perturbation screens. Imaging-based phenotyping efforts such as Cell Painting and high-content screening similarly seek to characterize cellular responses to chemical and genetic perturbations. Recent work found that deep image representations often perform on par with standard baselines, while batch effects and data-source variation remain important challenges for robust transfer (Borowa et al., 2024). Thus, the tension between scale, representation learning, and generalization appears to be a broader challenge for Virtual Cell models across modalities.

These findings motivate our central question: why does scaling fail for perturbation prediction when it succeeds elsewhere?

**Context-dependence as the missing factor.** A natural hypothesis is that the answer lies in biological context-dependence. Statistical genetics established decades ago that genetic effects are tissue- and context-specific: GTEx revealed pervasive tissue-specific eQTLs (GTEx Consortium, 2020), and TWAS studies show tissue-dependent genetic architectures (Gusev et al., 2016). At single-cell resolution, the same perturbation produces different transcriptional programs depending on cell state (Dixit et al., 2016; Frangieh et al., 2021). We argue this well-established property directly explains the generalization failures observed in perturbation prediction benchmarks.

**Connections to causal transportability.** Let $X$ denote the current cellular state, $P$ an experimentally applied perturbation, and $Y$ the resulting perturbed cellular profile. Standard distribution shift frameworks often assume that the predictive rule is stable across domains: the input distribution may change, e.g., $P_{\mathrm{src}}(X) \neq P_{\mathrm{tgt}}(X)$, while the conditional relationship remains fixed, e.g., $P_{\mathrm{src}}(Y \mid X) = P_{\mathrm{tgt}}(Y \mid X)$ (Quiñonero-Candela et al., 2009). Our setting is harder because biological context can change the mechanism linking perturbation to response, so that $P_{\mathrm{src}}(Y \mid X, do(P)) \neq P_{\mathrm{tgt}}(Y \mid X, do(P))$, where $do(\cdot)$ denotes Pearl's intervention notation (Pearl, 2009). In this regime, additional source-domain data alone cannot identify the target-domain response without target-domain interventions or explicit invariance assumptions. This connects directly to causal transportability (Pearl & Bareinboim, 2011; Bareinboim & Pearl, 2016), which formalizes when causal knowledge transfers across domains with different mechanisms. We develop this connection more formally in Section 3.

# 3. Theoretical Framework

## 3.1. The Standard Formulation

The prevailing methodology for predictive Virtual Cell models imports directly from large-scale machine learning: aggregate data at scale, train high-capacity models, and fine-tune for downstream tasks. This approach is grounded in the empirical observation of "scaling laws": model performance improves predictably as a power-law function of dataset size and parameter count (Kaplan et al., 2020; Hoffmann et al., 2022). The implicit assumption is that biology obeys similar laws.

The mathematical formulation for perturbation prediction typically takes the form:

$$f : (x, p) \mapsto y \tag{1}$$

where $x \in \mathbb{R}^G$ is the initial cellular state (e.g., gene expression over $G$ genes), $p \in \mathcal{P}$ is the perturbation (e.g., CRISPR knockout, cytokine), and $y \in \mathbb{R}^G$ is the post-perturbation expression. Given enough $(x, p, y)$ tuples, the model should approximate $f$ with increasing accuracy.

## 3.2. The Context Problem

What this formulation obscures is the biological *context* $c$, which is often unobserved or only partially captured in expression data. Context includes the microenvironment (neighboring cells, soluble factors), tissue of residence, developmental and epigenetic history, activation state (quiescent vs. proliferating, naïve vs. memory), temporal dynamics (cell cycle stage, time since stimulation), and genetics (germline variants, somatic mutations). See Appendix B for additional examples.

When $c$ varies, the function $f$ itself changes:

$$f_c : (x, p) \mapsto y \tag{2}$$

where the subscript indicates that the perturbation-to-response mapping depends on context. Critically, context acts as an *effect modifier*: it determines not just initial baseline expression but *how* perturbations affect the cell. A $CD8^+$ T cell in a draining lymph node responds differently to PD-1 blockade than the same cell type infiltrating a solid tumor (Miller et al., 2019). A hepatocyte in healthy liver metabolizes a drug differently than one in fibrotic tissue (Halpern et al., 2017).

By treating $c$ as implicit, we ask the model to recover context-dependent mechanisms from partially observed data. This is a problem that is not identifiable in general without additional assumptions.

## 3.3. Why Implicit Context Inference Fails

One might hope that the expression profile $x$ encodes sufficient information about $c$ for implicit inference. This is unlikely to succeed for three reasons:

**(1) Unobservable modifiers.** The transcriptome contains information about cellular context, including cell type and activation state, but it is likely to miss many causal drivers of perturbation response. Critical effect modifiers such as chromatin accessibility, protein abundance and localization, post-translational modifications, metabolite abundances, and rare donor-specific genetic variants are not directly observed in RNA-seq. As a result, two cells with similar baseline transcriptomes may differ in hidden regulatory or biochemical states that lead to different responses to the same perturbation.

**(2) Identifiability.** Even when contextual information is partially encoded in $x$, most current perturbation datasets cover only a limited set of contexts, with sparse overlap across contexts and perturbations. This makes it difficult to uniquely disentangle perturbation effects from context effects. Multiple context-dependent mechanisms may ex-

plain the same observed training data, especially when a perturbation is observed in only a small subset of contexts, leaving the target-context response underdetermined.

**(3) Causal non-invariance.** Predictive signals for context in RNA expression do not guarantee that the underlying perturbation-response mechanism remains invariant across domains. Context is not merely a nuisance covariate: it can determine which regulatory pathways, chromatin programs, or signaling states mediate the effect of a perturbation. Thus, even if a model infers a useful context label from $x$, the causal mechanism associated with that signal may shift in an unseen target domain, making implicit context inference unreliable for out-of-distribution transport.

### 3.4. Connection to Causal Transportability

We consider this problem through the lens of *causal transportability* (Pearl & Bareinboim, 2011; Bareinboim & Pearl, 2016), which studies when causal knowledge learned in one domain can be applied to another. The core question is: given interventional data from source domain $\pi$, under what assumptions can we identify the causal effect in a target domain $\pi^*$?

We define the causal perturbation effect within context $c$ as:

$$\delta(p; c) = \mathbb{E}[Y \mid do(P = p), C = c]$$
$$- \mathbb{E}[Y \mid do(P = \text{ctrl}), C = c] \quad (3)$$

where $P$ and $C$ are random variables representing perturbation and cellular context respectively, and $do(P = p)$ denotes an external intervention setting the perturbation to $p$. This represents the context-specific causal effect of perturbation $p$ (vs. control) on gene expression in context $c$, a conditional average treatment effect (CATE) that depends on context.

**Definition 3.1** (Informal). A causal effect is *transportable* from source domain $\pi$ to target domain $\pi^*$ if the target-domain causal effect can be computed from interventional data in $\pi$ and observational data in $\pi^*$, under stated assumptions about which mechanisms remain stable versus which differ across domains (Pearl & Bareinboim, 2011; Bareinboim & Pearl, 2016).

For cellular perturbation prediction, different biological contexts (e.g., cell types, tissues, activation states) correspond to different domains. In the transportability framework, changes in context across domains are represented through selection diagrams that indicate which causal mechanisms differ between $\pi$ and $\pi^*$ (Bareinboim & Pearl, 2016). When the mechanisms linking perturbation to response are among those that change, the causal effect generally cannot be transported without interventional data from the target domain.

**Proposition 3.2** (Informal). *When changes in context mod-*

*ulate the mechanism linking perturbation $P$ to response $Y$, predictions learned in source domain $\pi$ need not transfer to target domain $\pi^*$ with different contexts, without additional invariance assumptions or target-domain interventions.*

This perspective explains why accumulating more data from the same contexts yields diminishing returns: it refines performance for the training distribution of contexts, but does not inform the context-specific mechanism $f_{c^*}$ in a new target context $c^* \notin \text{supp}(P_\pi(C))$. In other words, we can learn $f_c$ well for contexts $c$ in our training, yet remain unable to predict $f_{c^*}$ for unseen contexts when the perturbation–response mechanism shifts in $c^*$.

### 3.5. Contrast with Standard Distribution Shift

It is tempting to view this transportability challenge through the lens of standard distribution shift, which the machine learning literature has extensively studied (Quiñonero-Candela et al., 2009). However, our setting differs fundamentally from standard covariate shift. In covariate shift, the input distribution changes, e.g., $P_{\text{src}}(X) \neq P_{\text{tgt}}(X)$, while the predictive rule is assumed stable, e.g., $P_{\text{src}}(Y \mid X) = P_{\text{tgt}}(Y \mid X)$. Similarly, domain adaptation methods typically rely on shared conditional structure across domains with shifted covariates (Ben-David et al., 2010).

In perturbation prediction, the relevant mechanism can change with biological context: $P_{\text{src}}(Y \mid X, do(P)) \neq P_{\text{tgt}}(Y \mid X, do(P))$. This represents a mechanism shift rather than merely a change in the distribution of inputs. Thus, methods designed for covariate shift, such as importance weighting or domain-adversarial training, do not directly address the failure mode. When the underlying perturbation-response mechanism changes, additional source-domain data alone cannot recover the target mechanism without target-domain interventions or explicit invariance assumptions.

This places perturbation prediction in a different regime. Addressing this challenge requires either (1) sufficient contextual diversity in training data to observe perturbation effects across the range of mechanisms encountered at test time, or (2) explicit modeling of how context modulates response mechanisms.

## 4. Empirical Evidence

Our theoretical framework predicts that contextual diversity, not just data volume, drives cross-context generalization. We evaluate this prediction through published benchmarks and controlled analysis of a large-scale immunology dataset.

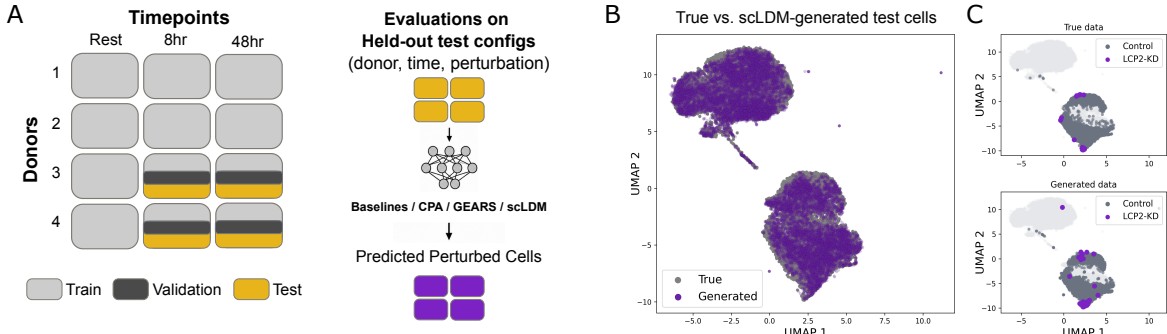

*Figure 2.* **Cross-context generalization task.** (A) Dataset split for evaluating generalization across contexts. Configuration is defined as a (Donor, Timepoint, Perturbation) triplet. Test configurations are held out for evaluation. (B) UMAP visualization of true and scLDM-generated cells for 25 representative test configurations. (C) Example distribution shift under LCP2 knockdown (KD): true cells (top) versus scLDM-generated cells (bottom), showing that scLDM captures the shift.

## 4.1. Simple Baselines Match Deep Learning

If perturbation prediction followed typical scaling laws, we would expect steady improvement as models grow. The evidence suggests otherwise. Ahlmann-Eltze et al. (2025) systematically compared single cell foundation models (scGPT (Cui et al., 2024), scFoundation (Hao et al., 2024), Geneformer (Theodoris et al., 2023), scBERT (Yang et al., 2022), UCE (Rosen et al., 2023)) and deep learning perturbation methods (GEARS (Roohani et al., 2022), CPA (Lotfollahi et al., 2023)) against simple baselines. The results were stark:

**No deep learning method consistently outperformed baselines**. For combinatorial perturbations on the Norman dataset (Norman et al., 2019), a simple additive baseline achieved lower error than all deep models (Ahlmann-Eltze et al., 2025). For single perturbations, no deep learning model consistently outperformed a mean-prediction baseline (Ahlmann-Eltze et al., 2025; Li et al., 2025b).

**Many models collapse to dataset-level averages.** Ahlmann-Eltze et al. (2025) observed that predictions from scGPT, UCE, and scBERT *varied little across perturbations* while GEARS and scFoundation varied considerably less than ground truth. These findings align with Wu et al. (2024), who found that models achieving reasonable fit-based scores (RMSE, cosine similarity) can fail completely on rank metrics, revealing "mode collapse". Li et al. (2025b) characterizes this as "variance compression," where fine-tuning collapses pretrained variability into overly smooth outputs that fail to capture the true diversity of perturbation.

Why do models *appear* to achieve reasonable-seeming performance despite not learning perturbation-specific effects? Viñas Torné et al. (2025) identified the root cause as *systematic variation*: consistent transcriptional shifts between perturbed and control cells shared across perturbations within a dataset. Systematic variation allows a model to score well by simply predicting the average response. Viñas Torné et al.

(2025) found that systematic variation strongly correlates with model performance: $r = 0.91$ for fine-tuned scGPT and $r = 0.95$ for GEARS. Standard metrics thus overestimate generalization by conflating systematic variation with perturbation-specific prediction.

Our theoretical framework offers an explanation for this pattern: narrow context coverage may bias models toward learning context-specific correlations rather than transportable mechanisms.

## 4.2. A Controlled Test of Context versus Scale

Published benchmarks have established that scaling has not improved perturbation prediction, but they do not isolate *why*. To directly test whether context diversity or data volume drives generalization, we analyze a genome-scale Perturb-seq dataset of primary human CD4$^+$ T cells (Zhu et al., 2025). This dataset consists of 22 million cells, approximately 12,000 gene knockdowns, four donors, and three activation timepoints (resting, 8hr, 48hr post-stimulation). This dataset provides diversity along two context axes while holding cell type fixed. Here, *context* is operationalized as donor identity crossed with activation timepoint, yielding up to 12 distinct biological contexts. Donor variation reflects genetic differences that modulate perturbation responses through regulatory variants; activation timepoint captures cell state dynamics. T cells are particularly informative because their response to the same perturbation varies dramatically with activation state (Miller et al., 2019), precisely the mechanism shift our theoretical framework describes.

**Task and Methods.** We construct a cross-context transfer task in which models train on perturbations from a subset of donor-timepoint combinations and predict effects in held-out contexts (Figure 2A). Eligible test perturbations are restricted to those observed in training contexts, so the task isolates seen perturbations in unseen contexts rather than

generalization to novel perturbation identities. We benchmark a state-of-the-art model, scLDM (Palla et al., 2025), along with existing deep perturbation-prediction models CPA (Lotfollahi et al., 2023) and GEARS (Roohani et al., 2022), on this task against simple baselines. We further use the benchmark as a controlled setting for identifying which properties of the data support cross-context generalization. Full methodology appears in Appendix C. The question motivating our analysis is not which model is the best, but *what data properties enable generalization when it occurs*.

**Perturbation Effect (Δ).** Under randomized perturbation assignment within each context $c = (\text{donor} = d, \text{timepoint} = t)$, the causal effect defined in (3) can be identified as:

$$\delta_{p;d,t} = \mathbb{E}[Y \mid P = p, d, t] - \mathbb{E}[Y \mid P = \text{ctrl}, d, t] \quad (4)$$

which is equivalent to the difference between the averaged (i.e. pseudo-bulk) expression of perturbed and control cells, i.e., $\Delta_{p,d,t}$, a commonly used effect summary (Wei et al., 2025; Li et al., 2025b; Viñas Torné et al., 2025).

**Metrics.** We evaluate two complementary aspects of performance. First, aggregate metrics that measure overall similarity between predicted and true perturbation effects Δ, including Pearson correlation, cosine distance, L2 distance, as well as a rank-based discrimination score. Second, DEG-F1 (F1 score of DEG recovery) measures whether predictions recover the correct differentially expressed genes (DEGs), i.e., genes whose expression changes under perturbation. DEGs are a primary output used to generate mechanistic hypotheses and nominate therapeutic targets; a model that matches aggregate statistics but misses key DEGs has limited practical utility. Additional details are provided in Appendix C.

### 4.3. Central Finding: Context Diversity Drives Generalization

**Aggregate metrics suggest measurable transfer in at least one model.** To track which data properties enable generalization, we first verify whether any model achieves measurable cross-context transfer. Qualitative assessments of generated cells show good overall reconstruction for scLDM (Figure 2B) and capture the distribution shift for a representative perturbation example (Figure 2C). By conventional standards, scLDM performs well on held-out contexts, outperforming mean baselines on Δ-based correlation, cosine distance, and L2 distance (Figure 3A–C). The gap between scLDM and simple baselines is narrower when using Spearman correlation-Δ, a rank-based nonlinear association metric, suggesting that some aggregate improvements depend on the choice of Δ-based score (See Appendix C, Figure C2). Discrimination scores are consistently high (me-

dian > 0.9) across held-out donor-timepoint combinations (Figure 3D, and Figure C1) and exceed baseline performance. CPA and GEARS do not consistently outperform simple baselines across aggregate metrics, consistent with recent benchmarks showing that deep perturbation models often fail to dominate simpler approaches (Ahlmann-Eltze et al., 2025). These results suggest that with explicit context conditioning, some cross-context transfer is achievable in T cells, motivating our analysis of which data factors drive this transfer.

**DEG recovery reveals biological limitations.** However, recovery of differentially expressed genes tells a different story. For scLDM, precision is reasonable (median ≈ 0.67), meaning that predicted DEGs are usually correct. But recall is poor (median ≈ 0.09), meaning the model recovers only ∼9% of true DEGs. For a biologist designing follow-up experiments, this implies ∼91% of the relevant biology is missed. We also evaluated CPA, which showed low DEG recovery (average precision and recall < 0.02).

Critically, DEG-F1 shows only limited association with aggregate metrics (Spearman correlation $r = 0.38$ and $r = -0.13$ with Correlation-Δ and L2-Δ, respectively; Figure 3E–F). Many configurations achieve Correlation-Δ $\geq 0.7$ yet DEG-F1 $< 0.1$, and the discordance is especially pronounced for L2-Δ. This dissociation suggests that standard evaluation practices may overestimate biological relevance. A model can appear successful by aggregate measures while failing at the task that matters for downstream applications.

**Generalization improves with context coverage, not just scale.** This is our central empirical finding. For each test configuration $(d, t, p)$, we quantified two properties of the training data: (i) the number of distinct donor-timepoint contexts in which $p$ was observed, and (ii) the total number of training cells with perturbation $p$.

DEG-F1 improves substantially with context coverage (mean DEG-F1 increases from $\leq 0.1$ to $\sim 0.19$ when moving from $\leq 3$ to 8 contexts; Figure 4A), indicating that perturbations observed across more training contexts yield better biological recovery in held-out contexts (a weaker trend was observed for L2-Δ, Figure 4B). In contrast, DEG-F1 shows only a weak association with cell count (Spearman's $r = 0.10$; Figure 4C). Notably, aggregate metric L2-Δ correlates more strongly with cell count (Spearman's $r = -0.52$; Figure 4D), suggesting that scale can improve standard scores even when biologically meaningful DEG recovery does not necessarily improve. We observed qualitatively similar trends for CPA, although its overall DEG-F1 is low (see Appendix C, Figure C3).

To control for cell counts, we compared configurations whose perturbation was observed in a similar number of

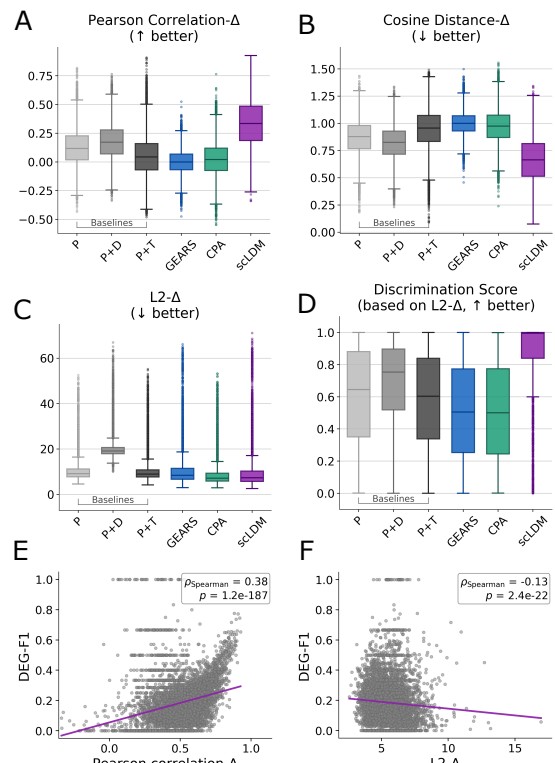

*Figure 3.* **Aggregate metrics suggest successful transfer in at least one model**. (A–C) scLDM outperforms mean baselines on Δ-based metrics, including Pearson correlation, cosine distance, and L2 distance, and generally exceeds CPA and GEARS, except for L2-Δ where CPA is competitive. GEARS was evaluated on the subset of test perturbations it could support. (D) L2-Δ-based discrimination scores for one held-out context show strong perturbation ranking for scLDM (median > 0.9; other contexts shown in Figure C1). (E–F) DEG-F1 correlates weakly with Δ-based metrics, especially L2-Δ, highlighting a dissociation between standard benchmarks and biological recovery.

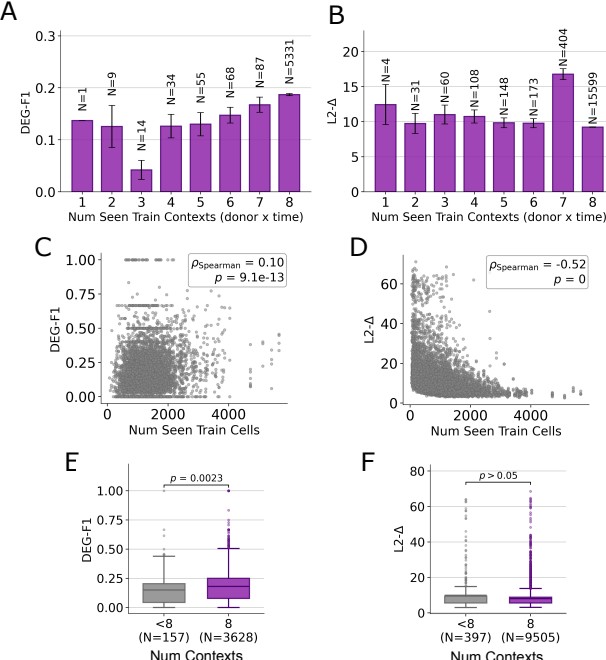

*Figure 4.* **Context diversity drives cross-context DEG recovery.** (A–B) DEG-F1, and to a lesser extent L2-Δ, improves with the number of training contexts per perturbation. $N$ denotes the number of test configurations $(d, t, p)$ in each group. DEG analysis was run on the subset of test configurations containing at least 100 perturbed cells. (C–D) DEG-F1 correlates weakly with training cell count (Spearman's $r = 0.10$), whereas L2-Δ correlates more strongly (Spearman's $r = -0.52$), suggesting scale can improve aggregate scores without consistent gains in biological recovery. (E–F) A controlled comparison of test configurations with similar training cell counts (700–1700) but low (< 8) versus high (8) context coverage shows the benefit of broader context exposure. $N$ denotes the number of test configurations in each group. Statistical significance was assessed using a one-sided Mann–Whitney U test.

training cells (700–1700). The high-diversity group (8 training contexts) significantly outperformed the low-diversity group (< 8 contexts) for DEG-F1 ($p = 0.002$), while the difference in L2-Δ was not significant ($p \geq 0.05$). *Even at controlled data volume*, context diversity improves generalization of biological recovery.

### 4.4. Interpretation

These findings directly support our transportability analysis (§3). When biological context modulates the perturbation-response mechanism, as it does for T cell activation states, observing more cells from the *same* context refines estimates of that context-specific mechanism but provides limited information about held-out contexts. This is analogous to extensively characterizing drug responses in naïve T cells while expecting predictions to transfer to exhausted T cells in the tumor microenvironment, where the relevant regula-

tory logic differs (Miller et al., 2019).

By contrast, observing the same perturbation across diverse contexts (donors, activation states) allows models to disentangle context-invariant effects from context-specific modulations. This is precisely the knowledge required for cross-context generalization.

The practical implication for the field is clear. **A dataset with 1 million cells across 100 contexts can be more valuable than 10 million cells across 10 contexts.** The main barrier to Virtual Cells is often not insufficient model capacity or cell counts, but datasets that sample contexts too narrowly to support causal transport across the biological contexts where predictions must apply.

## 5. Alternative Views

We address four credible counterarguments.

**1. "Scale will eventually solve this."** Large language mod-

els have shown strong few-shot in-context learning with scale, and multiple lines of work report emergent abilities that appear only beyond certain scales (Brown et al., 2020; Wei et al., 2022). Perhaps cellular foundation models are simply pre-emergent: context-awareness will spontaneously arise once models are large enough and trained on sufficient data. The current failures reflect insufficient scale, not fundamental limitations.

*Our response.* This argument deserves serious consideration because emergence has genuinely surprised researchers in NLP. However, we identify two reasons for the skepticism.

First, scaling succeeds when the target function is *context-invariant*. AlphaFold (Jumper et al., 2021) achieves near-experimental accuracy because the mapping from sequence to native structure is largely governed by universal physical constraints (Anfinsen, 1973). While intracellular conditions and chaperones can influence folding kinetics and proteostasis, the native fold of a given sequence is far less context-dependent than cellular perturbation responses, which can change substantially across states and environments.

Second, language contains explicit context signals. When GPT-style models handle context-dependent meaning, they can exploit textual cues: surrounding sentences, discourse markers, and speaker or document structure. Training data contains billions of examples where context is linguistically marked. In contrast, single-cell transcriptomes typically lack explicit context markers for many relevant variables, and the observed transcriptome can conflate multiple latent factors. We remain open to surprises, but see limited mechanistic basis for expecting scaling alone to reliably recover unseen context-specific response mechanisms.

**2. "Context could be inferred from transcriptomes."** The transcriptome is not random noise. It reflects the cell's state, including aspects of context. Indeed, cell type classifiers can achieve high accuracy from gene expression alone. With sufficient model capacity, a foundation model should learn to infer context and condition predictions accordingly. Explicit context labels are unnecessary; the information is present in the data.

*Our response.* We agree that expression encodes contextual information and that models can learn cell type representations. The question is whether this suffices for perturbation prediction. Three factors suggest not: (1) the transcriptome is a lossy projection of cell state: cells with similar RNA profiles can differ in chromatin state and respond differently to perturbation; (2) the inference problem is confounded: expression changes with context, so using expression alone to infer context is ambiguous because different hidden factors can produce similar expression profiles; (3) current models demonstrably fail: Ahlmann-Eltze et al. (2025) found that scGPT, UCE, and scBERT predictions often do not vary appropriately across perturbations, instead predicting context-averaged outputs.

**3. "Current benchmarks misrepresent foundation model capabilities."** Foundation models are designed for representation learning, not direct perturbation prediction. The right evaluation is downstream transfer: do embeddings improve tasks such as cell type annotation, batch correction, or other cell state analyses? On these tasks, foundation models often show clear benefits (Cui et al., 2024; Liu et al., 2024).

*Our response.* We do not dispute that foundation models help discriminative tasks. Cell type annotation and batch correction require recognizing patterns that may be context-invariant (e.g., "T cells express the marker gene CD3 regardless of activation state"). Perturbation prediction is fundamentally different: it is generative and counterfactual, requiring prediction of what *would happen* under intervention. This depends on context-specific mechanisms that standard discriminative benchmarks do not probe.

**4. "The field is young; give it time."** Protein structure prediction struggled for decades before AlphaFold. Early neural language models were long considered uncompetitive. Paradigm shifts take time. Single-cell foundation models are only a few years old.

*Our response.* We are sympathetic to this view and want to be clear: we argue for course correction, not abandonment. The rapid progress in single-cell modeling is real, and it is plausible that larger datasets and better inductive biases will eventually unlock stronger generalization. However, a key distinction applies. In protein folding, decades of incremental improvements produced steadily improving trajectories, suggesting that scaling and refinement were compounding. In contrast, recent perturbation-prediction benchmarks report mixed progress: despite substantial increases in data, model size, and architectural sophistication, performance has not consistently improved beyond simple baselines under standard evaluations (Ahlmann-Eltze et al., 2025; Viñas Torné et al., 2025; Wong et al., 2025).

## 6. Call to Action

Based on our analysis, we suggest three directions for the community.

**For data generation.** Our findings suggest that contextual diversity matters as much as cell count for context generalization (if not more). When designing perturbation screens, prioritizing coverage across donors, cell types, activation states, and timepoints may yield greater returns than depth within a few contexts. Under a fixed sequencing budget, there is an inherent tradeoff between broad context coverage and cell depth per context, and effective dataset design must balance breadth and depth.

A useful future benchmark would directly test this breadth vs. depth tradeoff under a matched cell-count budget: one training scheme would increase contextual breadth, while another would increase depth within fewer contexts. Large-scale datasets, such as Tahoe-100 (Gandhi et al., 2025), provide a natural setting for such experiments.

**For benchmarking.** Most perturbation benchmarks primarily evaluate generalization to *held-out perturbations* and/or *held-out combinations*, often within a narrow or fixed biological context (Ahlmann-Eltze et al., 2025; Li et al., 2025a). Fewer studies explicitly stress-test *cross-context transport* across covariates such as donors, timepoints, or cell types (Wei et al., 2025; Wu et al., 2024; Li et al., 2025a). We argue these settings probe complementary capabilities and should be reported separately, defining benchmarks with increasing difficulty: new perturbations, new contexts, and new perturbations in new contexts. Our empirical analysis focuses on the second setting (seen perturbations in new contexts), and evaluating unseen perturbations in either seen or new contexts remains an important direction for future work. Our results suggest that strong $\Delta$-based scores can mask biologically meaningful failures, motivating multi-metric evaluations that also include DEG recovery. Appendix D reviews the pros and cons of different metrics.

**For modeling.** Our results suggest that making context explicit, rather than relying on models to infer it from expression alone, may be necessary for cross-context generalization. This could involve conditioning on observed context covariates (e.g. cell type, activation state, tissue, donor) or using in-context learning approaches where sets of cells serve as prompts defining biological context at inference time (Dong et al., 2026). Causal representation learning under sparse mechanism shifts can separate stable factors from context- or perturbation-specific changes, improving generalization (Lopez et al., 2023). Finally, in many applications, context-specialized models (e.g. a Virtual CD4$^+$ T cell) may be more actionable than a universal Virtual Cell.

## 7. Discussion and Conclusion

**Isn't context-dependence obvious?** Biologists have understood context-dependence for decades. Our contribution is not rediscovering this fact but synthesizing fragmented evidence, providing theoretical grounding via transportability, and articulating why the AlphaFold analogy fails for predicting cellular perturbations. One narrative in the field remains that we are "one scaling breakthrough away" from Virtual Cells. The evidence suggests otherwise.

**Are we saying Virtual Cells are impossible?** No. We argue that the current approach, which scales data and model capacity within narrow context distributions, is misaligned with the problem structure. Virtual Cells may require ei-

ther substantially greater contextual diversity in the training data, explicit modeling of how context modulates response mechanisms, or a shift toward context-specialized rather than universal models; each of these paths is viable.

**Limitations.** Our empirical analysis focuses on transcriptomic readouts; whether conclusions extend to proteomics, imaging-based phenotyping, and other modalities remains an important direction for future investigation. Our theoretical framework assumes context is discrete and annotatable, whereas real biological contexts often exist on a continuum. We do not address how much contextual diversity is "enough," as this likely depends on the target application.

When explicit context labels are absent, causal representation learning offers a complementary path for inferring latent effect modifiers. For example, sparse mechanism shift models treat perturbations as interventions on an unknown sparse subset of latent variables, encouraging representations that separate biological processes affected by perturbation and improve out-of-domain transfer (Lopez et al., 2023). However, such disentanglement is generally underdetermined from transcriptomics alone; practical progress will likely require richer multimodal measurements that expose effect modifiers not visible in RNA, together with stronger biological priors.

**Conclusion.** The vision of a universal Virtual Cell has captured the imagination of both machine learning and biology. Yet the route to this goal is unlikely to mirror the scaling playbook that succeeded in protein folding or language. The central obstacle is biological context: cellular responses to perturbations vary across cell state, microenvironment, and history. Consequently, the target is a family of context-indexed mechanisms, and additional data from the same contexts yields diminishing returns. We support this view with evidence from recent benchmarks and new experiments showing that context coverage, and not just cell count, drives cross-context generalization. Additionally, we connect these observations to mechanism shift and causal transportability, clarifying why naive scaling can be insufficient.

The Virtual Cell remains a worthy aspiration. Achieving it will require respecting the biological reality that context matters.

## Code and Data Availability

The code for the scLDM model used in this study, along with the trained model parameters, is available at `https://github.com/czbiohub-chi/scldm_cd4`.
The CD4+ T cell Perturb-seq dataset used for model training and evaluation is available at `https://virtualcellmodels.cziscience.com/dataset/genome-scale-tcell-perturb-seq`.

## Acknowledgements

We thank Kavita Kulkarni, Amanda Surya, Giovanni Palla, Jakub Tomczak, Weimin Wu, and Dennis Wu for helpful discussions. A.A.K. was supported in part by NIH DP2AI177884, a Chan Zuckerberg Investigator Award, and a Breakthrough T1D/LRA/NMSS Common Mechanisms of Autoimmunity Insight Award (1-SRA-2025-1755-A-N). H.L. was supported by NIH R01LM1372201, NSF AST-2421845, Simons Foundation MPS-AI-00010513, and a Chan Zuckerberg Biohub Chicago Spoke Award. M.K. was supported by NIH T32GM139782.

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

# A. Model Descriptions

We provide a brief review of some popular deep learning methods trained on observational and perturbation single-cell transcriptomic data.

**Observational data models.**

scVI (Lopez et al., 2018) is an early deep probabilistic approach that uses variational inference with fully connected encoder-decoder components to model scRNA-seq count data, learning latent representations that account for batch effects and library size.

Geneformer (Theodoris et al., 2023) pretrains a Transformer encoder using rank-based value encoding and a BERT-style masked modeling objective. Genes are ranked by expression within each cell, and the model learns to predict masked genes from context.

scGPT (Cui et al., 2024) uses an autoregressive formulation on gene expression profiles, defining an implicit gene prediction order via a specialized attention mask and confidence-based iterative generation.

TranscriptFormer (Pearce et al., 2025) models gene-count tuples autoregressively as "cell sentences," conditioning each tuple on preceding tokens via log-likelihood training.

scBERT (Yang et al., 2022) and UCE (Rosen et al., 2023) use masking-based self-supervised approaches to learn transferable cell representations from large observational corpora.

**Perturbation data models.**

CPA (Lotfollahi et al., 2023) factorizes perturbation and covariate effects in a latent space to support extrapolation across doses, time points, and covariate settings.

GEARS (Roohani et al., 2022) integrates deep learning with a knowledge graph of gene-gene relationships via GNN-based inductive bias, enabling prediction for perturbations of genes not seen during training.

STATE (Adduri et al., 2025) combines an embedding model trained on observational data with a transformer-based transition model trained on perturbed cells, explicitly pairing covariate-matched control/perturbed sets.

CellFlow (Klein et al., 2025) models perturbation as a condition-guided continuous flow (neural ODE) that transports control-cell embeddings to the perturbed distribution, trained via flow matching with optimal-transport-based pairing of control and perturbed cells.

scLDM (Palla et al., 2025) first trains a transformer-based autoencoder with permutation-invariant/equivariant cross-attention to encode cell profiles into a latent space, then fits a latent diffusion prior trained with flow-matching (linear interpolants) and uses classifier-free guidance for controlled sampling under joint conditions. scLDM frames perturbation prediction as conditional *distribution* modeling rather than point prediction from matched controls. Specifically, it learns $P(X \mid p, c)$ for perturbation $p$ and context $c$, enabling counterfactual analysis of the distribution shift $P(X \mid \mathrm{ctrl}, c) \rightarrow P(X \mid p, c)$ and downstream *in-silico perturbation* analyses (e.g., DEGs recovery and perturbation ranking). In contrast to methods that rely on explicit cell-level control–perturbed pairings or anchor sets, scLDM can be trained without constructing matched pseudo-pairs.

# B. Extended Context Examples

To illustrate the scope of biological context, we provide examples of how context modulates perturbation response.

**Activation state.**   The same cytokine can produce proliferation in resting T cells but apoptosis in chronically activated T cells. PD-1 blockade has minimal effect on naive T cells but reinvigorates exhausted T cells in tumors (Miller et al., 2019). These differences arise because activation rewires signaling networks, altering which transcription factors are available and which chromatin regions are accessible.

**Tissue microenvironment.**   Hepatocytes in healthy liver metabolize drugs differently than hepatocytes in fibrotic tissue (Halpern et al., 2017). The same genetic perturbation in a T cell produces different outcomes depending on whether the cell

resides in a lymph node, circulates in blood, or infiltrates a solid tumor.

**Genetic background.** Expression quantitative trait loci (eQTLs) show that the same regulatory variant can have strong effects in one tissue and no effect in another (GTEx Consortium, 2020). Donor-to-donor variation in our T cell experiments likely reflects such genetic differences in regulatory wiring.

**Temporal dynamics.** Perturbation responses evolve over time. A knockdown measured at 8 hours post-stimulation captures different biology than the same knockdown at 48 hours, as cells progress through activation programs and the relevant signaling networks change.

These examples underscore that context is not a nuisance variable to be controlled away, but a fundamental determinant of biological response that models must account for.

## C. Full Methodology

### C.1. Dataset

The dataset from Zhu et al. (2025) comprises Perturb-seq measurements from primary human CD4$^+$ T cells with: (i) four healthy donors, providing biological replication; (ii) three activation timepoints (resting, 8hr, 48hr post-stimulation), capturing T cell activation dynamics; and (iii) approximately 12,000 single-gene CRISPRi knockdowns, providing dense genomic coverage.

### C.2. Data Preprocessing and Quality Control

Quality control was performed separately for each donor-timepoint combination. Only cells with a single guide assignment were retained. Cells expressing fewer than 100 genes were removed. Cells with abnormally low or high library sizes were excluded using Median Absolute Deviation (MAD) based outlier detection on total UMI counts, with minimum threshold 1,400 UMIs and MAD multiplier 9:

$$\max(1400, \text{median} - 9 \times \text{MAD}) \leq \text{counts} \leq \text{median} + 9 \times \text{MAD}$$

Cells were not filtered by mitochondrial read proportions; fewer than 2% of cells exceeded 20% mitochondrial reads, and mitochondrial expression was retained to capture stress-related perturbation responses.

Gene-level filtering removed genes expressed in fewer than 100 cells or with total count below 100. Highly variable genes (HVGs) were identified using Seurat's variance-stabilization method after library-size normalization to 10,000 counts and log-transformation. HVG selection was performed independently per timepoint using combined data from all donors, retaining 2,000 HVGs per timepoint. The union across timepoints yielded 3,699 genes for downstream analysis.

### C.3. Perturbation Filtering

We computed two knockdown metrics within each donor-timepoint combination:

- **Perturbation knockdown ratio:** Mean expression of target gene across perturbed cells divided by mean in controls

- **Cell knockdown ratio:** Expression of target gene in each perturbed cell divided by mean in controls

Perturbations were retained if their perturbation knockdown ratio was $< 0.5$ (at least 50% reduction) in at least two donors per timepoint. Cell-level filtering retained only cells with knockdown ratio $< 0.5$. Perturbations with fewer than 100 remaining cells across all conditions were removed. After filtering: approximately 18 million cells across 10,570 perturbations remained.

### C.4. Configuration Definition and Data Split

We define a *configuration* as a unique (Donor, Timepoint, Perturbation) triplet, allowing separate analysis of context effects and perturbation identity.

Held-out contexts comprised two donors at 8 hr and 48 hr timepoints. Within each held-out context, 30% of eligible perturbations were assigned to validation and 40% to test; the remaining perturbations were included in training, providing some in-context supervision. Eligible perturbations were restricted to those also observed in training contexts, ensuring the evaluation reflects context transfer rather than novel perturbations. Non-targeting controls were excluded from validation/test and retained only in training.

Final split: 100,125 training, 7,449 validation, and 16,527 test configurations, comprising 14,662,411, 1,084,349, and 2,423,397 cells, respectively.

### C.5. scLDM Architecture, Parameters and Training

Following the two-stage training procedure of scLDM (Palla et al., 2025), we trained (i) the autoencoder (AE) on training-set mRNA counts to learn a latent space, then (ii) the latent flow-matching component with *joint* conditioning on donor, timepoint, and perturbation. We used the validation set for model selection and hyperparameter tuning, monitoring AE reconstruction loss, flow-matching loss, and Correlation-$\Delta$ of predicted perturbation responses. Based on validation performance, we replaced the default count encoding with `softbin` encoding (10 bins), removed positional embeddings from latent tokens, and modeled the full HVG sequence without gene sampling. For the AE, we used model dimension 256, latent dimension 16, 4 self-attention and cross-attention heads, 256 inducing points, and 8 encoder/decoder layers. For the diffusion transformer (DiT), we used model dimension 512 with 8 layers and 8 attention heads, with *joint* conditioning on donor, timepoint, and perturbation embeddings. The AE was trained for 60 epochs on $8\times$ NVIDIA H100 GPUs, and the DiT was trained for 150 epochs on $32\times$ NVIDIA H100 GPUs; inference used $64\times$ NVIDIA H100 GPUs.

### C.6. Baseline Models

Following common practice in perturbation benchmarking (Ahlmann-Eltze et al., 2025), we constructed a perturbation-mean baseline (P) that, for each query configuration, outputs a single pseudo-bulk expression vector given by the average expression of training cells with the matched perturbation (or control). We further defined two context-matched mean baselines that additionally condition on donor (P+D) or timepoint (P+T), averaging over training cells matching the query configuration in perturbation and the corresponding context attribute. These baselines simply reflect dataset-level statistics; a successful learning approach should outperform them under cross-context evaluation.

In addition to these mean-based baselines, we compared against two neural perturbation-response models, GEARS (Roohani et al., 2022) and CPA (Lotfollahi et al., 2023). We trained both models following the protocols and hyperparameter settings described in their original papers. Specifically, we used the default model configurations, optimization parameters, and training procedures recommended by the authors, making only the minimal changes required to adapt the inputs and splits to our benchmark. This avoids additional baseline-specific tuning and provides a direct comparison to established perturbation prediction methods under the same cross-context evaluation setting.

### C.7. Evaluation Details

**UMAP visualizations.** We randomly selected 24 perturbed test configurations and one additional LCP2 knockdown configuration (chosen for its large true distribution shift; Figure 2C), together with their matched control cells. For computational efficiency, we subsampled cells from both perturbed and control groups. True and generated expression profiles were processed with library-size normalization (10,000 counts), `log1p` transformation, and feature scaling (`max_value`=10). We computed PCA (50 components), built a $k$-nearest neighbor graph ($k = 15$), and obtained UMAP embeddings (`min_dist`=0.3) using Scanpy (Wolf et al., 2018), fitting jointly on true and generated cells.

**$\Delta$-based metrics.** For each test configuration, we computed:

- **Correlation-$\Delta$:** Pearson correlation between predicted and true perturbation effects, $\Delta_{\text{pred}}$ and $\Delta_{\text{true}}$.

- **Cosine Distance-$\Delta$:** Cosine distance between predicted and true effects, defined as $1 - \cos(\Delta_{\text{pred}}, \Delta_{\text{true}})$.

- **L2-$\Delta$:** Euclidean distance between $\Delta_{\text{pred}}$ and $\Delta_{\text{true}}$.

We excluded 63 and 341 test configurations from $\Delta$-based metric calculations for the P+T and P+D mean baselines, respectively, because the required matching subsets (by perturbation + timepoint or perturbation + donor) were not present

in train split.

**DEGs computation.** Differentially expressed genes were computed for test configurations with at least 100 perturbed cells. For computational efficiency, we subsampled an equal number of cells from the corresponding context-matched controls. Using Scanpy (`scanpy.tl.rank_genes_groups`) (Wolf et al., 2018), we compared perturbed cells to matched controls via a Wilcoxon rank-sum test and defined DEGs as genes with Benjamini–Hochberg adjusted $p < 0.05$ and $|\log_2 \mathrm{FC}| > 0.25$. The same procedure was applied to both real and generated cells. Baseline models were excluded from DEG analysis since they produce a single pseudo-bulk profile rather than cell-resolved predictions.

**Statistical analysis.** Statistical significance in Figure 4E–F was assessed using one-sided Mann–Whitney U tests comparing test triplets whose perturbations were observed in the maximum number of training contexts (8 donor–timepoint combinations) versus those observed in fewer than 8. We tested whether maximal context exposure yields higher DEG-F1 and lower L2-$\Delta$.

**Discrimination score.** For each held-out context, we evaluated whether the model can correctly *rank* perturbations by how closely their predicted effects match the ground truth. We used L2-$\Delta$ (lower is better) and DEG-F1 (higher is better) as ranking criteria. Specifically, for each configuration with perturbation $p$, we ranked all candidate perturbations in that context by ascending L2-$\Delta$ (or descending DEG-F1) and recorded the rank of $p$. We report a normalized discrimination score, $1 - \frac{\mathrm{rank}(p)-1}{N}$, where $N$ is the number of candidate perturbations in that context (higher is better).

Results for one held-out context (Donor CE0010866, 8 hr) are shown in Figure 3D; results for the remaining held-out contexts are shown in Figure C1. scLDM outperforms all baselines across held-out contexts. In particular, DEG-F1-based discrimination is consistently harder than L2-$\Delta$-based discrimination (Figure C1), yielding lower scores. This aligns with our earlier observations that biologically grounded criteria such as DEG recovery expose failures that can be masked by aggregate $\Delta$-based metrics.

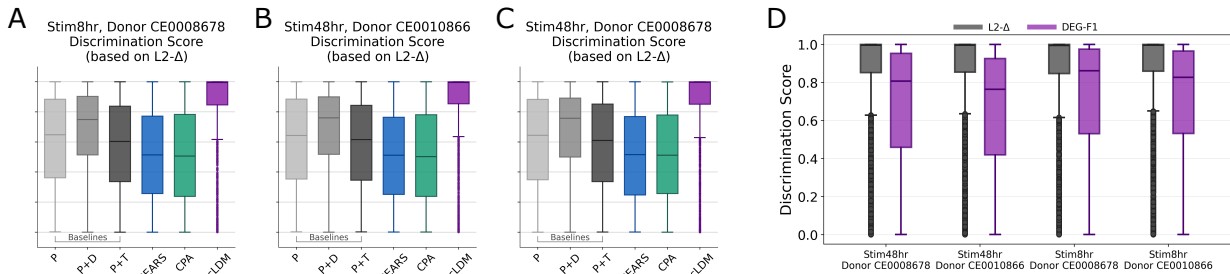

*Figure C1.* **Analysis of discrimination scores.** (A–C) L2-$\Delta$-based discrimination scores show that scLDM outperforms mean baselines and other tested methods in held-out contexts (median $> 0.9$), indicating accurate perturbation ranking. (D) DEG-F1-based discrimination scores are compared with L2-$\Delta$-based discrimination scores for scLDM. DEG-F1-based ranking is more biologically grounded but more challenging, yielding lower scores. The plot is limited to test configurations with at least 100 perturbed cells.

**Spearman's correlation-$\Delta$.** While Pearson correlation is more commonly used to evaluate $\Delta$-based effect summaries, Spearman's rank correlation does not assume a linear relationship between predicted and true effects. Under this rank-based metric, the gap between scLDM and simple baselines is narrower (Figure C2), suggesting that aggregate performance differences depend partly on the choice of $\Delta$-based score.

**CPA performance versus context and cell exposure.** CPA shows overall weak DEG recovery, but a qualitatively similar trend with context exposure as observed for scLDM: DEG-F1 increases with the number of training contexts per perturbation (Figure C3A). In contrast, L2-$\Delta$ does not show a meaningful change as context exposure increases (Figure C3B). The qualitative relationships between DEG-F1 and training cell count, as well as between L2-$\Delta$ and training cell count (Figure C3C–D), are also similar to those observed for scLDM.

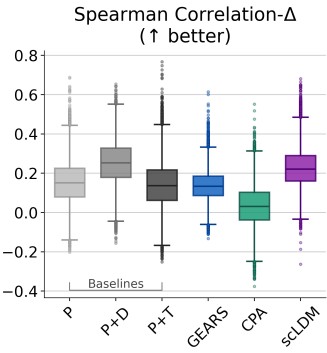

*Figure C2.* **Model comparison using Spearman correlation-Δ.** Spearman correlation-Δ provides a rank-based comparison between predicted and true perturbation effects. Under this metric, the gap between scLDM and simple baselines is narrower than under Pearson correlation-Δ.

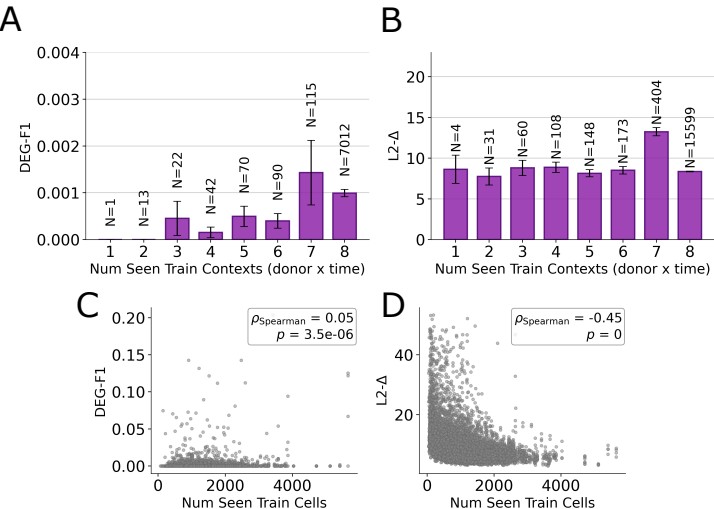

*Figure C3.* **CPA performance versus context and cell exposure.** (A–B) DEG-F1 shows a qualitative improving trend with the number of seen training contexts per perturbation, whereas L2-Δ does not. (C–D) DEG-F1 shows weak association with training cell count, whereas L2-Δ shows a stronger association with training cell count, qualitatively consistent with the trends observed for scLDM.

## D. Discussion of Evaluation Metrics

The choice of evaluation metric substantially affects conclusions about model performance. We discuss tradeoffs among commonly used metrics.

**Mean Absolute Error / Mean Squared Error.** MAE/MSE measure average reconstruction error in (normalized) gene expression, either across all genes or a subset such as HVGs. Their main advantages are simplicity and interpretability, but they can be sensitive to systematic variation (Viñas Torné et al., 2025), weight all genes equally regardless of biological relevance, and allow models to score well by regressing toward dataset-level averages. These metrics are also strongly influenced by measurement noise and dropout, especially for lowly expressed genes. This makes it hard to tell whether a zero value reflects true biological absence or a technical dropout event. In many settings, the resulting reconstruction error is on the same order as the typical per-gene perturbation effect size. As a result, aggregate MAE/MSE can be a weak proxy for accurate perturbation-effect prediction. Consistent with this, the Virtual Cell Challenge found that most models performed worse than a mean baseline on MAE, limiting its usefulness as a primary optimization target (Arc Institute, 2025).

**Correlation.** Correlation between true and generated gene expression values is another common metric for assessing overall reconstruction quality. In practice, models can achieve relatively high correlations, but this agreement is not necessarily indicative of accurate perturbation-effect prediction.

$\Delta$**-based metrics.** $\Delta$ summarizes the average perturbation effect as the difference between pseudo-bulk expression profiles of perturbed cells and their context-matched controls. Predicted and true $\Delta$ vectors are typically compared using Pearson correlation, cosine distance, or Euclidean distance. While widely used for perturbation benchmarking, $\Delta$-based metrics have two key limitations: (1) they ignore heterogeneity within perturbed and control populations (e.g., differences in variance or multimodality around the mean); and (2) they can be difficult to interpret biologically, since "desired effects" are defined in a normalized expression space rather than in terms of marker genes or DE programs. DEG-based evaluations therefore provide a more directly actionable complement.

**DEG recovery.** This measures overlap between true and predicted differentially expressed genes. Advantages include biological interpretability, alignment with experimental validation workflows, and focus on genes that matter for downstream applications. Disadvantages include sensitivity to threshold choices and requirement for sufficient cells to compute DEGs reliably.

**Perturbation discrimination.** This metric evaluates whether a model can distinguish perturbations by ranking their predicted effects against the ground truth. Its main advantage is that it directly tests perturbation specificity and aligns with downstream use cases such as prioritizing perturbations for experimental follow-up. An average discrimination score of $0.8$ implies that the true perturbation ranks, on average, within the top $\sim 20\%$ of candidates, potentially reducing the number of perturbations requiring follow-up by $\sim 80\%$. The limitations are that the score depends on the chosen distance metric and does not measure absolute accuracy: a model may rank perturbations correctly while still misestimating effect magnitudes.

**Our perspective.** We argue that DEG recovery deserves greater emphasis because it aligns closely with biological use cases: perturbation predictions are often used to prioritize experiments based on expected downstream transcriptional programs. A model can achieve high aggregate similarity yet fail to recover key DEGs, limiting its practical utility. In our experiments, DEG recovery is only weakly correlated with common aggregate metrics, indicating that these scores capture different aspects of performance. We therefore recommend reporting multiple metrics, including DEG-based evaluations, to provide a more complete picture of model capabilities.

