# OpenReview forum: "Position: Virtual Cells Need Context, Not Just Scale"
_ICML.cc/2026/Position_Paper_Track — ICML 2026 Position Paper Track regular_

### Official Review · Reviewer_THhm · 2026-02-26

**Significance:** 3
**Argument Clarity:** 3
**Rating:** 5
**Confidence:** 3

**Questions:**

**Questions**
1. Is pearson correlation the right correlation metric to be using here? Some of these relationships do not look linear.

**Alternative Views Section:**

Yes

**Compliance With Llm Reviewing Policy A Conservative:**

Affirmed.

**Discussion Potential:**

3

**Final Justification:**

I maintain my recommendation, with the insufficiency of the pearson correlation and legibility of the charts addressed I think this a strong position paper.

**Paper Summary:**

Recent negative results have shown virtual cell models unable to outperform simple baselines. Attempts to fix this by scaling model size and training data have so far been disappointing. The authors argue that this is because cellular processes, unlike for example, protein folding, are highly context sensitive. So collecting more data from the same contexts wont help virtual cells generalise. They show this empirically by trying to predict perturbations in T cells on held-out contexts with scLDM. They propose collecting data from cells in a greater number of contexts to tackle this, benchmarks which directly test for cross-contextual generalisation, and making context explicit in the modelling process.

**Position:**

Yes

**Position In Title:**

Yes

**Related Work:**

3

**Strengths And Weaknesses:**

**Strengths**
- The inclusion of an experiment that tries to demonstrate this effect strengthens the argument made considerably.
- I like the format of the alternative views section, following each objection with the authors response, and I felt it was thorough, preempting some of my own questions.
- I think given recent interest in virtual cells arising from the ARC virtual cell challenge and the controversy surrounding it, that this is a timely and thought provoking position paper.

**Weaknesses**
- Font size and overall plot size on some of the plots (figure 2C and 3 especially) is too small to read clearly.
- Some of the plots are unclear more generally, what does N refer to in 4E-F for instance?

**Support:**

4

---

> ### Author Rebuttal · Authors · 2026-03-31
>
> We sincerely thank Reviewer THhm for the strong support and for recognizing the timeliness, empirical grounding, and structural clarity of our position paper. We have addressed your main concerns below:
>
> **1. The Choice of Correlation Metric (Pearson vs. Non-Linearity)**
>
> >Is pearson correlation the right correlation metric to be using here? Some of these relationships do not look linear.
>
> This is a well-taken point. Some of the scatter plots (Figures 3E–F and 4C–D) do exhibit non-linear relationships for which Pearson correlation can be misleading. We made two changes in response:
>
> * **Action 1 (Scatter Plots):** We have replaced Pearson correlation with **Spearman rank correlation** in these figures. While the qualitative trends remain the same, Spearman provides a much more statistically appropriate and clearer summary of these relationships.
>
> * **Action 2 (New Perturbation Metric):** Motivated by this comment, we also evaluated Spearman-Δ as a perturbation-response metric alongside the Pearson-based Corr-Δ used in prior work and in our original submission. We recognized that evaluating the predicted perturbation effect vectors ($\Delta$) with Pearson, while standard in prior work, emphasizes linear agreement.. Therefore, we added **Spearman-$\Delta$** as a new evaluation metric to our main results. Under Spearman-$\Delta$, scLDM remains the strongest deep model, but the simple baselines become even more competitive. We have added a discussion of this to the text, as it perfectly reinforces our thesis about the importance of metric choice and suggests that aggregate correlation-based metrics may not fully capture biologically relevant performance such as DEG recovery.
>
> **2. Figure Legibility and Clarity**
> >Font size and overall plot size on some of the plots (figure 2C and 3 especially) is too small to read clearly.
>
> We agree that several panels were too small to read comfortably. In the revised manuscript we have fully reformatted Figures 2, 3, and 4 for the camera-ready version. We increased the base font sizes, enlarged the individual panels (especially the distribution shifts in 2C and the boxplots in 3), and reduced the point sizes in the scatter plots to minimize overplotting.
>
> **3. Clarifying Notation in Figures 4E-F**
> > Some of the plots are unclear more generally, what does N refer to in 4E-F for instance?
>
> Thank you for catching this ambiguity. We have updated the main text and explicitly revised the caption for Figure 4 to state that **$N$ denotes the number of unique test configurations** (specifically, the held-out Donor, Timepoint, Perturbation triplets) included in each respective group.
>
> ---
>
> We hope these metric refinements and visual overhauls fully address your concerns. We are deeply grateful for your advocacy of this paper and hope these revisions solidify your support for acceptance.

---

> > ### Author Rebuttal · Reviewer_THhm · 2026-04-02
> >
> > Thank you for the thoughtful and considered rebuttal and promised improvements.
> >
> > My main concerns were with the suitability of the pearson correlation and legibility of the plots. These are quite easily fixed and the authors endeavour to do so.

---

### Official Review · Reviewer_gfkX · 2026-03-11

**Significance:** 3
**Argument Clarity:** 4
**Rating:** 6
**Confidence:** 5

**Questions:**

Can authors put the context of virtual cell beyond genetic perturbations and include other "virtual cell" attempts. E.g. in CellPainting and HCS or limit the work explicitely to genetic perturbations and RNASeq or other data modalities?

**Alternative Views Section:**

Yes

**Compliance With Llm Reviewing Policy A Conservative:**

Affirmed.

**Discussion Potential:**

4

**Final Justification:**

Rebuttal addressed my concerns.

**Paper Summary:**

The paper argues that development of virtual cell is not about scaling up, but more about contextualizing the information that goes to the model. There is a broad discussion showcasing that simple linear baselines outperforms multiple foundational models. The paper introduces how biology matters behind the perturbation data, as different cell lines, anatomical locations, and cell types or organism influence the foundational models development and their utility, namely underpromise. The call to action is to go directly into context awareness and better incorporation of causality and biology inforrmation into to the models to improve their utility.

**Position:**

Yes

**Position In Title:**

Yes

**Related Work:**

2

**Strengths And Weaknesses:**

the position is clearly stated, it is very actionable and very well backed and argumented with evidence from the analysis and recent benchmarks

There is a strong clear reasoning process behind the work, showcasing exactly why biology context matters and cannot be omited. Also there is alternative view to that, fairly described.

The topic is from interdisciplinary research, as ICML community in recent years tries to build foundational models for biology, it is important but its impact may go beyond this community.

Definitely the work will spark discussion whether scale will not solve the problem of underperforming of foundational models, but also how to incorporate some biological knowledge and mechanisms into modeling, likely results in completely different family of models.

Regarding the related works, I would add results from cell painting / high content imaging community, as even there where variability or context is vastly reduced, still foundational methods did not surpassed the baselines based on CellProfiler as presented in [1].

[1] Borowa, Adriana, et al. "Decoding phenotypic screening: A comparative analysis of image representations." Computational and Structural Biotechnology Journal 23 (2024): 1181-1188.

**Support:**

4

---

> ### Author Rebuttal · Authors · 2026-03-31
>
> We sincerely thank Reviewer gfkX for the strong support and the constructive suggestion to bridge our argument with the high-content imaging community. We address this directly below.
>
> **1. Extending the "Virtual Cell" Context to Cell Painting and HCS**
> > Can authors put the context of virtual cell beyond genetic perturbations... I would add results from cell painting / high content imaging community, as presented in Borowa et al. (2024).
>
> We completely agree. The tension between scaling representation learning and achieving robust biological generalization is remarkably consistent across modalities. We think this is a structural issue in AI-for-biology, not a modality-specific quirk.
>
> We added a new paragraph after the discussion of empirical challenges to the scaling paradigm in Related Work (Section 2). Also, We revised the scope statement to explicitly name imaging-based modalities and proteomics as critical next frontiers in Limitations (Section 7).
>
> We believe these additions provide a much richer, cross-disciplinary perspective to our Call to Action. We are grateful for your suggestions to this paper and your constructive guidance.

---

> > ### Author Rebuttal · Reviewer_gfkX · 2026-04-02
> >
> > All was answered

---

### Official Review · Reviewer_Xcbt · 2026-03-13

**Significance:** 3
**Argument Clarity:** 3
**Rating:** 4
**Confidence:** 4

**Questions:**

1. The empirical analysis in Section 4 relies solely on scLDM. Would the conclusion be equally significant if a foundation model with a different inductive bias (like scGPT or Geneformer) were used? It would be highly beneficial to add one or two comparison models.
2. Given that explicit context labels might not always be fully annotated or observable in observational datasets, how do you see the role of unsupervised causal representation learning in practically disentangling context from perturbation effects moving forward?

**Alternative Views Section:**

Yes

**Compliance With Llm Reviewing Policy A Conservative:**

Affirmed.

**Discussion Potential:**

2

**Final Justification:**

I thank the authors for their response in the rebuttal period. I have no further questions and will maintain my positive score.

**Paper Summary:**

This paper argues that scaling model capacity is insufficient to solve the Virtual Cell problem. Within a given biological context, simple baselines perform on par with sophisticated model architectures, and current models fail to consistently generalize across different contexts. The authors substantiate their argument through the analysis of a state-of-the-art model on a 22-million-cell immunology dataset. The experiments further demonstrate that contextual diversity and causal representation learning deserve increased emphasis, complementing the ongoing scaling of model capacity and data volume.

**Position:**

Yes

**Position In Title:**

Yes

**Related Work:**

2

**Strengths And Weaknesses:**

Strengths
1. Rigorously mapping the biological concept of 'Context' to the machine learning concepts of 'Causal Transportability' and 'Mechanism Shift' provides a highly compelling theoretical explanation for the poor generalization capabilities of single-cell foundation models.
2. The experimental design is reasonable. Rather than merely benchmarking numerous models to achieve SOTA, the authors utilize a 22M-cell dataset to decouple the impact of data scale from contextual diversity by holding the cell type constant while varying donor identities and activation timepoints.
3. The writing is well-structured with clear contributions. The authors explicitly list three core contributions, including theoretical formalization, an empirical analysis of a 22-million-cell dataset, and concrete recommendations. The entire paper is tightly organized around these three points.
Weaknesses：
1. The empirical analysis relies heavily on a single model (scLDM) and lacks validation across foundation models with different inductive biases (such as scGPT or Geneformer). The persuasiveness of the empirical evidence leaves room for improvement.
2. While the authors argue against the notion that context can be implicitly inferred from the transcriptome, their argumentation for this dismissal remains underdeveloped.

**Support:**

3

---

> ### Author Rebuttal · Authors · 2026-03-31
>
> We sincerely thank Reviewer Xcbt for the encouraging feedback and for highlighting the strength of our theoretical mapping and experimental design. Prompted by your feedback, we have run new experiments and refined our text. We address your specific points below:
>
> **1. Generalization Across Different Inductive Biases**
>
> >The empirical analysis relies solely on scLDM. Would the conclusion hold for models with different inductive biases like scGPT or Geneformer?
>
> We agree that demonstrating this context barrier across multiple model families strengthens the paper's core thesis. Retraining massive, pretrained foundation models like scGPT or Geneformer from scratch in a controlled, split-matched setting is unfortunately not computationally feasible within a short rebuttal window.
>
> However, to directly answer your underlying question—whether different inductive biases can overcome the context barrier—**we have added new experiments evaluating GEARS and CPA**. These established perturbation-response models utilize entirely different inductive biases from scLDM (graph neural networks and latent covariate factorization, respectively).
>
> * **The Results:** Both GEARS and CPA consistently underperform simple mean baselines on Correlation-Δ and Cosine Distance-Δ.
> * **The Scaling Trend Holds:** Because GEARS performed near random on average in held-out contexts, we focused our DEG analysis on CPA. While CPA's absolute DEG-F1 is lower than scLDM's, we observe the **similar qualitative scaling trend**: CPA's average DEG recovery improves with the *number of seen training contexts*, while showing a relatively weak correlation with the total number of seen cells.
> * **Action:** We have added these multi-model results to the main text and appendix. This suggests that the observed scaling with context, as well as the relatively poor DEG recovery under limited context exposure, are not specific to scLDM and may persist across different architectures.
>
> **2. The Transcriptome as a "Lossy" proxy (Section 3.3)**
>
> >The argumentation dismissing implicit context inference from the transcriptome remains underdeveloped.
>
> We apologize our text sounded dismissive; we do not seek to claim that context is absent from RNA expression. Rather, our point is that RNA expression is a **lossy proxy** for the causal drivers of perturbation response, making implicit context inference unreliable for out-of-distribution transport.
>
> We have revised and softened Section 3.3 to explicitly detail three specific bottlenecks:
> * **Unobservable Modifiers:** Critical effect modifiers—such as chromatin accessibility, protein state, or rare donor-specific genetic variants—can fundamentally alter a cell's response mechanism without leaving a uniquely identifiable *baseline* transcriptomic signature.
> * **Identifiability:** Even when an RNA signal exists, if most perturbations are only seen in a small subset of contexts, the training data lacks the crossed perturbation-context observations necessary to mathematically disentangle the context effect from the perturbation effect.
> * **Non-Invariance:** Finding a predictive signal for context in RNA does not guarantee that the causal mechanism linked to that signal remains invariant across unseen target domains.
>
> **3. The Role of Unsupervised Causal Representation Learning**
>
> >Given explicit labels might not always be available, how can unsupervised causal representation learning practically disentangle context moving forward?
>
> Thanks for this great question. We agree that in observational datasets, explicit context labels are often incomplete. We view unsupervised causal representation learning not as a replacement for explicit context, but as a critical tool to infer latent factors that better capture these causal effect modifiers.
>
> However, we argue that this disentanglement is incredibly difficult to achieve from transcriptomic data alone without additional structural assumptions. Practically moving forward, we see this succeeding through:
> * **Multimodal Integration:** Using foundation models that integrate epigenomic, proteomic, and spatial data, which capture the physical effect modifiers that RNA misses.
> * **Biologically Motivated Inductive Biases:** Designing objective functions that explicitly encourage the separation of baseline state, context, and perturbation-specific effects.
>
> **Action:** We have added a paragraph to the Discussion (Section 7) addressing this directly, clarifying that latent context inference is a highly valuable future direction, but its success will likely depend on richer, multi-modal data and stronger structural causal assumptions rather than scale alone.
>
> ---
>
> We hope these new experiments and theoretical refinements fully address your concerns. We believe your feedback has made this a substantially stronger position paper.

---

> > ### Author Rebuttal · Reviewer_Xcbt · 2026-04-03
> >
> > The authors have addressed all concerns.

---

### Official Review · Reviewer_Fqh5 · 2026-03-16

**Significance:** 3
**Argument Clarity:** 2
**Rating:** 3
**Confidence:** 4

**Questions:**

- In section 2 what is $P$?
- How are the results different when testing novel perturbations in existing contexts versus testing seen perturbations in unseen contexts?

**Alternative Views Section:**

Yes

**Compliance With Llm Reviewing Policy A Conservative:**

Affirmed.

**Discussion Potential:**

2

**Final Justification:**

The authors and I seem to converge on the fact that this is a useful framing but could benefit from additional empirical experiments demonstrating support. My score remains the same.

**Paper Summary:**

The authors of this paper argue that, in the pursuit of virtual cell models, the actual blocker in terms of well-performing models is not capacity, as what's been prioritized so far, including data capacity and model capacity, as well as model architecture innovation. Instead, it is genomic context. They draw a parallel between context-invariant problems such as protein folding and instead argue that causal transport literature is more relevant as a theoretical underpinning for the problems at hand. They present experiments on a 22 million cell CD4 perturbseq dataset. They are able to identify differentially expressed genes, and this performance improves with the number of donor time points and instead correlates weakly with cell count.

**Position:**

Yes

**Position In Title:**

Yes

**Related Work:**

2

**Strengths And Weaknesses:**

### Strengths
- The paper is timely as there are multiple ongoing challenges and dataset generation efforts
- The strongest part of the publication is the experimental evidence presented in high context diversity vs low context diversity
- Alphafold comparison is a strong point of the paper

### Weaknesses
- The experimental data demonstrated in the paper is quite weak. The central claim is tested on one dataset with a narrow range of context axes 3 time points and 4 donors.
- The Tahoe 100 dataset provides perturbation data across 50 cell lines, offering a wider range of genomic contexts
- Would the authors observe additive performance gains from scaling the number of cell lines in Tahoe-100?
- The connection to causal transport in section 2 feels underdeveloped. What is $P$? It appears without clear definition
- the authors argue that mechanism shift (where $P(Y|X,P)$ changes across contexts) is fundamentally different from standard covariate shift. I don't see why this is at its core different
- The argument that gene expression cannot capture context 3.3 is the weakest part of the paper. But donor identity, activation state, and cell type are not independent of gene expression.
- Do the authors note the performance of TxPert, STATE, or Tahoe-X1? For a position paper arguing about the general failure mode of the field, showing these methods also fail to overcome the context barrier would strengthen the case.

Some minor comments
- Figure 2 overplotting
- Figure 2 is there a way to demonstrate this without relying on UMAPs?

**Support:**

2

---

> ### Author Rebuttal · Authors · 2026-03-31
>
> We sincerely thank Reviewer Fqh5 for their rigorous and constructive review. We are particularly glad you found the AlphaFold comparison strong and our core experiments on contextual diversity compelling. Prompted by your feedback, we have conducted new experiments and significantly refined our theoretical definitions to strengthen the manuscript.
>
> Here is a detailed response to your points:
>
> **1. Evaluating Additional Models (TxPert, STATE, Tahoe-X1) & Different Inductive Biases**
>
> >Showing these methods also fail to overcome the context barrier would strengthen the case.
>
> We agree that demonstrating the context barrier across multiple model families strengthens our position. Fully retraining massive foundation models like STATE or Tahoe-X1 in a controlled, split-matched setting is unfortunately not computationally feasible within the rebuttal window.
>
> However, to directly address your underlying question—whether different models face the same context barrier—we have added new experiments evaluating GEARS and CPA. These established architectures have entirely different inductive biases from scLDM (ie GEARS = graph neural networks and CPA = latent covariate factorization).
>
> Our new results show that both GEARS and CPA consistently underperform simple baselines on at least two of the $\Delta$-based metrics. Also, when evaluating DEG recovery for CPA, we observed relatively weak performance compared to our main model, scLDM. Despite poorer DEG recovery, we observe similar qualitative scaling trend as we did for scLDM: average DEG-F1 improves  with the *number of seen training contexts*, and correlates relatively weakly with the total number of seen training cells. We have added these multi-model results to the manuscript, showing that the main empirical trends persist across multiple architectures with different inductive biases.
>
> **2. Mechanism Shift vs. Covariate Shift & Causal Transport**
>
> >Connection to causal transport in Section 2 is underdeveloped. What is P? I don't see why mechanism shift is at its core different from standard covariate shift.
>
> We apologize for the compressed notation and have revised this section. We explicitly define $P$ as the Perturbation and $C$ as the Biological Context. The core difference lies in *what* changes across domains. In standard covariate shift, the input distribution changes $P_{src}(X) \neq P_{tgt}(X)$, but the predictive rules remain identical $P_{src}(Y|X) = P_{tgt}(Y|X)$. Under mechanism shift, the biological "rules of the game" physically change across contexts: $P_{src}(Y|do(P), C) \neq P_{tgt}(Y|do(P), C)$. Accumulating infinite source data cannot identify the target mechanism without explicit causal assumptions or target-domain interventions. We have made this mathematical distinction more clear in Section 3.5.
>
> **3. The Transcriptome as a "Lossy" Context (Section 3.3)**
>
> >The argument that gene expression cannot capture context is weak. Donor identity, activation state, etc., are not independent of gene expression.
>
> We agree with your critique that donor identity, activation state, and cell type are not independent of gene expression. Our text unintentionally overstated this. Our refined argument in Section 3.3 is that RNA expression is a *lossy proxy* for the true causal drivers of perturbation response. We clarify three specific bottlenecks:
>
> - **Unobservable modifiers:** Chromatin state or rare genetic variants can alter response mechanisms without leaving uniquely identifiable baseline RNA signatures.
> - **Identifiability:** Limited perturbation-context overlap makes it mathematically difficult to uniquely disentangle context effects from perturbation effects.
> - **Non-Invariance:** Predictive RNA signals do not guarantee cross-domain mechanism invariance.
>
> **4. Novel Perturbations vs. Unseen Contexts**
>
> >How are results different when testing novel perturbations in existing contexts versus seen perturbations in unseen contexts?
>
> Our paper exclusively evaluates *seen perturbations in unseen contexts*. Evaluating novel perturbations is a fundamentally different task requiring additional machinery for out-of-vocabulary perturbation representation. We have revised the discussion to explicitly scope our claims to cross-context transport, highlighting the simultaneous test of "novel perturbations in unseen contexts" as the ultimate future benchmark.
>
> **5. UMAPs and Overplotting (Figure 2)**
> Thank you. We agree regarding Figure 2. We will revise the figure to reduce overplotting and  increase font sizes. Importantly, the UMAPs are purely qualitative visual aids; our empirical claims rely entirely on the robust quantitative evaluations in Figures 3 and 4.
>
> ---
>
> We hope these additions and theoretical clarifications fully address your concerns. We believe your feedback has substantially elevated the rigor of this position paper.

---

> > ### Author Rebuttal · Reviewer_Fqh5 · 2026-04-02
> >
> > Thank you to the authors for their additional explanation. It clarifies some of my concerns
> >
> > There is still missing evidence that would result in a clear acceptance from my perspective. I find that the evidence that additional context allows models to generalize better is still limited. There do exist biological datasets such as Tahoe 100 that make this a testable hypothesis.

---

### Decision · Program_Chairs · 2026-04-30

**Decision:**

Accept (regular)

**Comment:**

This position paper challenges the dominant assumption in the Virtual Cell field that training ever-larger models on more single-cell data will eventually solve the problem. The core argument is that the real bottleneck is not model capacity but a lack of diversity in biological contexts, supported by the observation that simple baselines match sophisticated models within a given context, while all current models fail to generalize reliably across contexts. Drawing on the causal inference literature on transportability, the authors conclude that the community faces a causal transportability problem and argue that contextual diversity and causal representation learning should receive much more attention alongside ongoing scaling efforts. Overall, the reviewers lean towards accept (1x strong accept, 1x accept, 1x borderline accept, 1x borderline reject). They appreciate the idea of mapping the biological concept of 'Context' to the machine learning concepts of 'Causal Transportability' and 'Mechanism Shift'. Though the empirical evidence for the failure mode is just one dataset, in my opinion(as a position) this is strong and something that will raise a discussion.